# Configurable topological textures in strain graded ferroelectric nanoplates

Kwang-Eun Kim[1], Seuri Jeong[1], Kanghyun Chu[1], Jin Hong Lee [1], Gi-Yeop Kim[2,3], Fei Xue[4], Tae Yeong Koo[5], Long-Qing Chen[4], Si-Young Choi [2,6], Ramamoorthy Ramesh[7,8,9] & Chan-Ho Yang[1,10]

Topological defects in matter behave collectively to form highly non-trivial structures called topological textures that are characterised by conserved quantities such as the winding number. Here we show that an epitaxial ferroelectric square nanoplate of bismuth ferrite subjected to a large strain gradient (as much as $10^5 \, m^{-1}$) associated with misfit strain relaxation enables five discrete levels for the ferroelectric topological invariant of the entire system because of its peculiar radial quadrant domain texture and its inherent domain wall chirality. The total winding number of the topological texture can be configured from −1 to 3 by selective non-local electric switching of the quadrant domains. By using angle-resolved piezoresponse force microscopy in conjunction with local winding number analysis, we directly identify the existence of vortices and anti-vortices, observe pair creation and annihilation and manipulate the net number of vortices. Our findings offer a useful concept for multi-level topological defect memory.

[1] Department of Physics, Korea Advanced Institute of Science and Technology (KAIST), Yuseong-gu, Daejeon 34141, Republic of Korea. [2] Department of Materials Modelling and Characterization, Korea Institute of Materials Science, ChangwonGyeongnam, 51508, Republic of Korea. [3] Department of Materials Science and Engineering, Pusan National University, Geumjeong-gu, Busan 46241, Republic of Korea. [4] Department of Materials Science and Engineering, The Pennsylvania State University, University Park, PA 16802, USA. [5] Pohang Accelerator Laboratory, POSTECH, Pohang, Gyeongbuk 37673, Republic of Korea. [6] Department of Materials Science and Engineering, POSTECH, Pohang, Gyeongbuk 37673, Republic of Korea. [7] Department of Materials Science and Engineering, University of California, Berkeley, CA 94720, USA. [8] Department of Physics, University of California, Berkeley, CA 94720, USA. [9] Materials Sciences Division, Lawrence Berkeley National Laboratory, Berkeley, CA 94720, USA. [10] KAIST Institute for the NanoCentury, KAIST, Yuseong-gu, Daejeon 34141, Republic of Korea. Correspondence and requests for materials should be addressed to C.-H.Y. (email: chyang@kaist.ac.kr)

Topological defects are singularities such as vortices in real, momentum and complex-phase spaces[1–3]. Topological defects are unstable in terms of their own energy, but they are created by other constraints such as given boundary conditions and system-specific symmetry[4]. The topological concept has been broadly applied to a variety of subjects and has led to novel physical phenomena such as the quantum Hall effect and the topological insulator[5] based on the topology of the quantum mechanical wave function, i.e., the orbital degree of freedom in solids. Magnetic vortices and skyrmions are also topological objects that are relevant to the spin degree of freedom[6–8]. The remaining fundamental degree of freedom that has been relatively rarely explored concerns the lattice in crystalline solids. Despite the identification of electric vortex structures[9–21], electric switching of competing vortex textures with deterministic configurability of the topological number remains experimentally unconfirmed. Therefore, the study of configurable topological defects in electrically polarised media such as a ferroelectric presents an opportunity for a complete understanding of the universal topological features in matter.

In this study, we explore the role of an inhomogeneous strain field as a mechanism for topological defects in ferroelectrics. Elastically deformed lattice in an inhomogeneous strain state is coupled with the ferroelectric property through the mechano-electric effect[22–24]. Nevertheless, the strain-gradient-induced effect has been overlooked as an origin of topological ferroelectric textures, because the polarisation induced by macroscopic bending is negligible relative to the typical value of spontaneous ferroelectric polarisation. However, recent advances in nanoscale characterisation have led to the discovery that large strain gradients are often present in epitaxial films relaxed from misfit strains[23,24], self-assembled nanostructures[25], dislocations[26], domain and twin walls[2], and morphotropic phase boundaries[27,28]. The challenge at hand, therefore, is to demonstrate ferroelectric materials subjected to significantly large inhomogeneous strains to clamp non-trivial textures and facilitate inter-phase switching. Direct observation and analysis of electric vortices in the context of the topological winding number in such curved lattices can provide an unprecedented view of ferroelectrics.

In the following results, we study how to stabilise, observe and control the ferroelectric topological textures in the epitaxial square nanoplate of bismuth ferrite subjected to a large strain gradient. The piezoresponse vector map of the ferroelectric nanoplate is obtained by angle-resolved piezoresponse force microscopy (PFM) and the positions of the vortices and antivortices are determined by winding number calculation based on the piezoresponse vector map. In addition, the effect of inhomogeneous strain in the ferroelectric nanoplate on non-trivial topological texture formation is investigated by phase field simulation. Finally, we show that the total winding number of the topological texture can be modulated by selective domain switching.

## Results

**Emergence of a radial quadrant domain structure.** Self-assembled $BiFeO_3$ (BFO) nanoplates were synthesized by pulsed laser deposition using a composite target mixing BFO with cobalt-ferrite spinel (see Methods section for the details). BFO has a rhombohedral structure with a large spontaneous ferroelectric polarisation (almost $100 \, \mu C \, cm^{-2}$) along a pseudocubic <111> direction in bulk[29] and weakly strained epitaxial films[30,31]. The large-area topographic image (Fig. 1a) shows the emergence of protruding square BFO plates with a typical lateral size of approximately 300 nm. Out-of-plane (OOP) and in-plane (IP)

PFM images reveal that a quadrant domain structure emerges on the BFO nanoplates (Fig. 1b, c). This unusual domain structure is attributed to the anisotropic mechanical boundary condition, i.e., the bottom of the BFO nanoplate is compressively strained while the other side and top faces experience no external stress. Quantitative analysis of the strain relaxation using X-rays and theoretical understanding based on phase field simulation will be discussed later in this study.

In the as-grown state, outward/upward polarisations are stabilised in most quadrant domain areas, but slim buffer domains with inward/downward polarisations are identified in the form of a cross. The electric poling over a square region by a positively biased tip switches the upward polarisation to the downward direction (Fig. 1d). Interestingly, this OOP switching is accompanied by the reversal of IP piezoresponse. The clamping of OOP and IP components suggests that each quadrant is subject to ferroelastic elongation along <111> and the poling gives rise to 180° ferroelectric switching along the rhombohedral axis. The switched configuration can be reversed to the original upward/outward state, as demonstrated by the reverse poling (Fig. 1e). Similar quadrant textures are commonly observed in almost all nanoplates regardless of differences in the lateral size and shape, which indicates the stability of the quadrant domain textures.

**Detailed intra-structure of a quadrant domain texture.** One of the greatest challenges in ferroelectric defect studies is to devise a direct real-space detection technique that observes the behaviour of electric vortices with nanoscale spatial resolution in a non-destructive manner, particularly when related to mechanical deformation. Angle-resolved PFM was used to construct IP piezoresponse vector map (Fig. 2). The in-plane PFM technique can distinguish only the perpendicular component to the cantilever, because it relies on the torsional vibration mode of the cantilever. The in-plane piezoresponse vector can be determined by using several high-resolution PFM images acquired in a quasi-identical region with different tip orientation angles, respectively. We carried out trigonometric curve fitting to determine the amplitude and phase shift of the sinusoidal fit function per position and mapped out the piezoresponse vector spatially.

We successfully visualised the piezoresponse vector distribution, thereby disclosing detailed features on an emergent radial quadrant domain structure (Fig. 3). The second and fourth quadrant ferroelastic domains of the rhombohedral BFO were split into a quadrant domain and a thin buffer domain with forming a 180° charged domain wall, respectively. At the ferroelectric domain walls, not only does the polarisation rotate to avoid uncompensated charge density, but the amplitude of polarisation is reduced to avoid imposing a significant energy cost for rotating the polarisation away from the easy axis in the lattice[32]. According to the locations of the buffer domains on the second and fourth quadrants, the measured nanoplate corresponds to the type 2 configuration (as will be addressed later) that contains two vortices at the ferroelectric domain walls and a single antivortex at the centre. Scanning transmission electron microscopy (TEM) of a cross-section of a nanoplate underpins the outward/upward quadrant polarisations and the downward buffer domain (Supplementary Fig. 5). Although the PFM vector map provides a useful insight into ferroelectric domain structures under the assumption that piezoresponse vector has a linear correlation with electric polarisation, we should be cautiously aware of the limitation. For example, it has been reported that 180° domain walls can generate a lateral piezoreponse due to a topographical slope at the domain boundary caused by opposite

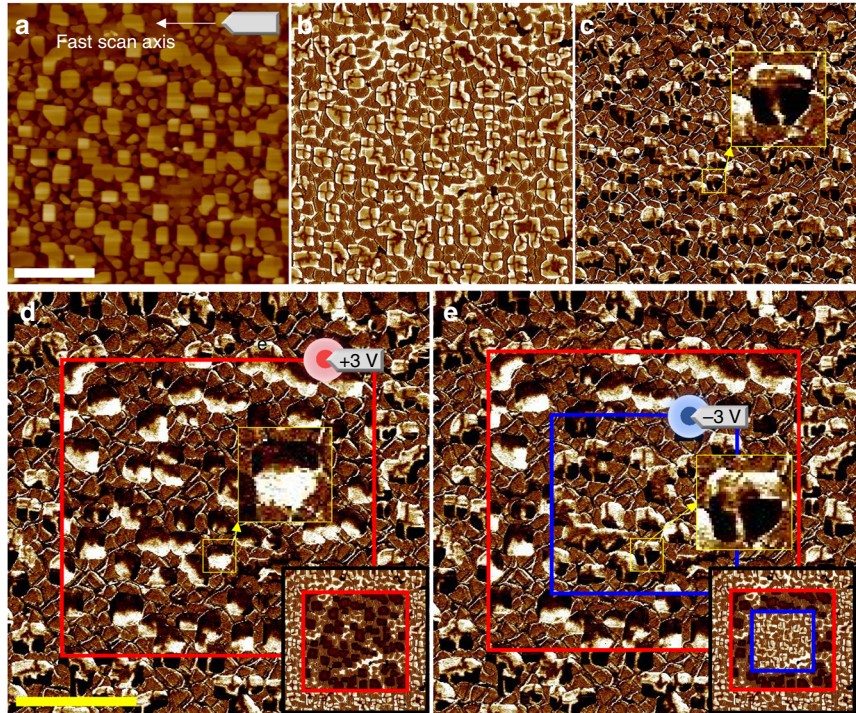

**Fig. 1** Large area PFM images of a double box switching region. **a** Surface topographic image. **b** OOP PFM image. **c** IP PFM image. All as-grown BFO nanoplates were observed to have an upward polarisation with the outward radial-quadrant domain structure. **d** IP PFM and OOP PFM (inset) images acquired after an electric poling. Inset: the upward polarisation in the as-grown state (bright contrast) was switched to a downward one (dark contrast) in the red box area that was scanned by a *dc* biased tip at +3 V. Simultaneously, IP PFM contrasts within the nanoplates were reversed by the poling, indicting a strong coupling between the OOP and IP polarisations. We note that various irregularly shaped nanoplates exhibit the similar radial-quadrant domain structure. **e** We attempted additional switching using a *dc* biased tip at −3 V inside the blue box to confirm the reversible nature of the switching. Scale bars represent 2 μm. The horizontal axes of images are parallel to the crystallographic peudocubic axis [100]

deformations on the neighbouring up and down polarised domains[33]. We are not sure how largely the effect is involved in our case, and thus, it is desirable to interpret the detailed feature of the domain walls based on theoretical supports through topological analysis and phase field simulation.

**Topological winding number analysis**. We analysed the measured vector map by calculating local winding numbers to identify the precise positions of the vortices (Fig. 3d). The winding number is a topological quantity that counts the singularities in vector fields and is defined in a two-dimensional space by contour integration of the variation of the vector direction along a given closed loop[1,2,34]; it is an integer that indicates the net number of singularities inside the loop, so the number is preserved in continuous deformations. The integral over a large enclosed space is equivalent to the total sum of all individual local winding numbers for small areas that comprise the large space (Supplementary Fig. 1). This conservation property ensures that the topological number of an entire system is determined only by the boundary condition and does not vary, regardless of any interior configuration.

We tiled small edge sharing loops and calculated local winding numbers (see the Methods section for the details). As a result, two vortices and a single antivortex were clearly identified. A single vortex point was found on each 180° charged domain wall. The anti-vortex was detected at the central merging point of the two-in/two-out domain configuration. The net sum of the vortex points in this as-grown state was + 1 and this net value was equal to the total winding number calculated along a large closed loop near the edge of BFO nanoplate. We emphasise that the

topological point affects not only the small loop area but also its far-field configuration globally; it can be easily verified that any other larger loops that only enclose a vortex result in the same winding number. Any random noise in the angle distribution from measurement artefacts and/or intrinsic incoherent fluctuations hardly destroys the robust topological nature.

**Phase field simulation**. Although the non-trivial topological texture consumes considerable energy in terms of mutual interactions among electric dipoles, their inevitable presence is due to a larger energy gain in another degree of freedom, namely the elastic energy. To reveal the origin of the radial quadrant domain structure, we calculate the inhomogeneous strain distribution in a nanoplate as illustrated in Fig. 4a by phase field simulations (see methods for details). As the bottom interface is constrained while the other five surfaces are stress-free, the mechanical boundary condition gives rise to the distribution of shear strain $\varepsilon_{xz}$ or $\varepsilon_{yz}$ as shown in Fig. 4b. Owing to the electrostrictive interaction $q_{1313}\varepsilon_{xz}P_xP_z$ and $q_{2323}\varepsilon_{yz}P_yP_z$ with $q_{ijkl}$ as the electrostrictive coefficients, the quadrant domains will be induced. After including the effect of depolarisation field, a domain pattern shown in Fig. 4c is obtained. The polarisation vectors on the top surface is demonstrated in Fig. 4d, which agrees well with the experimental observation in Fig. 3c. Therefore, the spontaneous rhombohedral deformation of BFO (rhombohedral angle ~ 0.6°)[35], combined with the specific distribution of shear strains caused by the mechanical boundary conditions of a nanoplate, leads to the formation of a quadrant ferroelastic domain structure in which each quadrant domain is elastically elongated along an outward <111> axis.

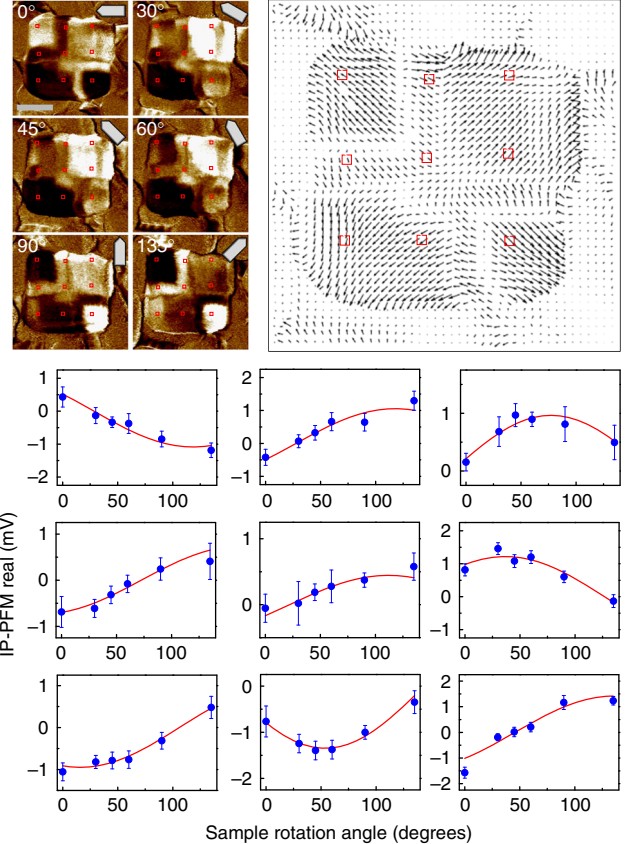

**Fig. 2** Construction of an IP piezoresponse vector map. IP PFM real part contrast images of a BFO nanoplate in the as-grown state of the type 2. As the IP PFM measurement can detect the perpendicular component to cantilever orientation, it is necessary to collect angle-dependent IP PFM signals at each position, in order to determine both the magnitude and direction of an IP piezoresponse vector. Angle-dependent IP PFM signals were fitted to a trigonometric curve at the representative positions marked by red rectangles. The amplitude and phase information of the fitting curve determine the amplitude and direction of the IP piezoresponse vector, respectively. A data point in the graph consists of an average value of $5 \times 5$ pixels in the PFM image and the error bar is defined by half of the difference between the maximum value and the minimum value within $5 \times 5$ pixels. The scale bar in the IP PFM image represents 200 nm

As shown in Supplementary Fig. 2a, the gradients of normal strains are largest at the four corners and the polarisations therein are enforced to be upward by the flexoelectric interactions[24,36]. Thus, the depolarisation field can only flip down the polarisation in the middle part, and the buffer domains are created in the vicinity of ferroelastic walls.

**Domain wall chirality and topological domain textures**. Under the elastic constraints, ferroelectric polarisation values are assigned to the quadrant domains. A ferroelastic quadrant domain can have two variant ferroelectric polarisations harmonised with each <111> elongation axis. In the as-grown state, upward/outward radial polarisations are stabilised and encounter electric frustration in the central region, thereby leading to non-trivial topological textures with a total winding number of + 1. A buffer ferroelectric domain with an inward/downward polarisation is built on one side of a ferroelastic domain wall to reduce the depolarisation energy, as indicated by the dark grey boxes in Fig. 5a. One end of the 180° domain wall between a buffer

domain and a quadrant domain is terminated at an edge where a strong strain gradient is present. The different strain states in both sides of the domain wall deviates the in-plane polarisation inter-angle between two neighbouring ferroelectric domains from 180°, pinning the domain wall chirality at the edge because IP polarisations in the domain wall should rotate gradually along an acute angle.

We note that a single (anti)vortex must exist at a point at which two different chiral domain walls are encountered. Although the nanoscale vortex itself is energetically unstable, the existence cannot be avoided topologically between two end points clamped to have mutually opposite chiralities. This vortex is expected to be readily movable between the two end points, potentially offering an isolated quasi-particle carrying energy along the one-dimensional chain. The observed vortex in the second quadrant (on the upper left) in Fig. 3d was located away from the symmetric position, suggesting a vortex formed by the domain wall chirality is less massive between the pinning ends and thus the location is vulnerable to influence from strain variations and/or uncontrolled perturbations.

The existence of buffer ferroelectric domains and the diversity of their locations create more abundance in possible topological textures (Fig. 5b, c). In the as-grown state in which ferroelectric polarisations in the quadrant domains are restricted outward (simultaneously upward), we can classify all 16 possible configurations into four types according to the locations of the buffer domains on one side of each red ferroelastic wall. However, the total winding number in the as-grown state is still + 1, because any buffer domain between two quadrant domains equally outward does not affect the total winding number (Supplementary Fig. 3). All the four types of buffer domain arrangements in the as-grown state can be stabilised by phase field simulations (Fig. 5b).

**Non-local domain switching and buffer domains control**. The central region of the BFO nanoplate can be electrostatically unstable due to the same polarity of the electric dipoles merging at the centre. Because of electrical frustration, unexpected complex domain structures are easily created alongside pair creations by an electrical writing at the centre of the nanoplate. To avoid touching the central region with the biased tip and minimise low-lying excitations, we used a non-local domain switching technique. When we apply a $dc$ bias to switch the electric polarisation on the corner of a ferroelastic domain, as described in Fig. 6a, the entire region of a ferroelastic domain is switched in addition to the electrically written area. In the nanoplate structure, the domain wall energy is comparable to the bulk energy because of the large surface-to-volume ratio. Therefore, the entire region of a ferroelastic domain is switched instead of creating a domain wall. This non-local switching offers a useful pathway into domain switching minimising artificial pair creations at the sensitive centre area leading to the minimum states. Figure 6b demonstrates our non-local domain switching process in a deterministic way. We applied $dc - 4$ V to the bottom electrode and scanned the corner part that was approximately 100 nm from the central area using a grounded tip. The electrical writing size was $400 \times 400$ nm, drawing 128 lines at a tip speed of 400 nm s$^{-1}$, including the corner region. We recognised the nanoplate region using the deflection error simultaneously acquired during the poling, which clearly indicated the boundary of the BFO nanoplate in real time. After non-local domain switching, PFM was measured to check the switching effect. We performed non-local domain switching at each corner of the BFO nanoplate one by one in a clockwise direction. Eventually, the entire ferroelastic domains were

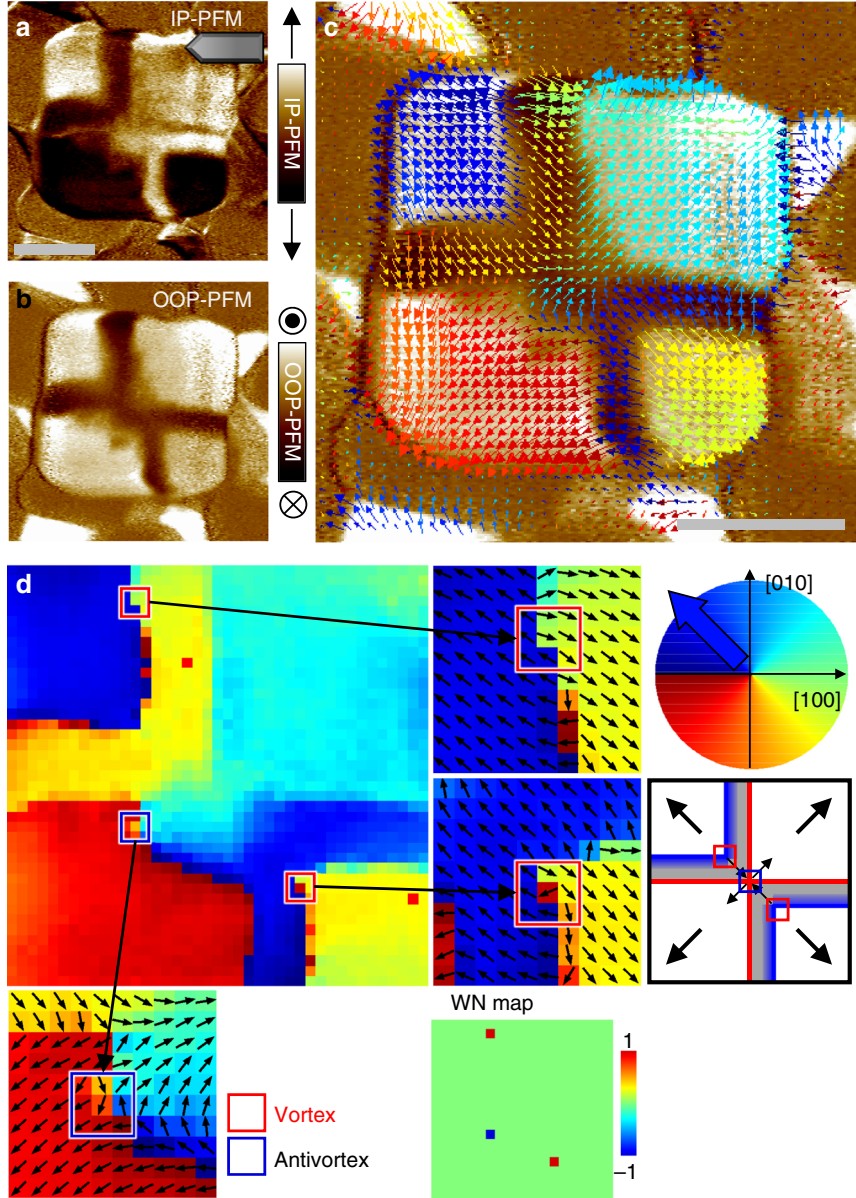

**Fig. 3** Observation of the vortex and antivortex points in a BFO nanoplate in an as-grown state by angle-resolved PFM measurements. **a** IP PFM image was measured at a tip orientation described by the illustration at the upper right corner. The bright (dark) contrast represents the IP piezoresponse vector component perpendicular to the tip orientation, i.e., pointing to the positive (negative) vertical direction. **b** OOP PFM image simultaneously measured. Most areas inside the plate region exhibit upward polarisation except for the buffer domains, in which a weak piezoresponse (shown in brown) was detected. **c** Map of local IP piezoresponse vectors. This map was constructed by combining the six IP PFM images measured with different tip orientation angles. Each colour arrow represents the direction of the piezoresponse vector as depicted in the coloured circle in **d**. The locations of the buffer domains indicate that this nanoplate corresponds to the type 2 configuration (Fig. 5c). **d** Colour map indicates the direction (angle) of IP piezoresponse vector. We tiled small-closed square (3 × 3) loops by overlapping their edges and calculated the winding number at each loop, thereby constructing the winding number map (WN map). Most areas appeared to have a winding number of 0, except for two vortices (red boxes; each + 1) and one antivortex (blue box; each − 1), as depicted in the schematic on the right-hand side. This as-grown domain configuration consists of upward polarisations in all quadrant domains, and topologically it results in a total winding number of 1. Scale bars represent 200 nm

switched to downward/inward polarisation from upward/outward polarisation.

Using non-local domain switching, we can control the existence or non-existence of the buffer domain and its location, which offers a useful pathway into manipulation of the total winding number. For example, we consider an initial domain structure with two neighbouring upward quadrant domains with a downward buffer domain placed on the left side of the ferroelastic wall between them (Fig. 6c). The right ferroelastic quadrant domain that does not contain the buffer domain is

switched first and the left ferroelastic domain is switched later. This sequence causes all relevant regions to have downward polarisations without creating a buffer domain. However, the reversal of the switching order leaves a buffer domain on the right side of the ferroelastic wall between quadrant domains that are otherwise the same. These multiple states can be understood on the basis of the competition between the depolarisation energy gain and the domain wall energy loss. The first domain switching causes positive and negative bound charges at the surface. In this case, the depolarisation energy is already stable without regard to

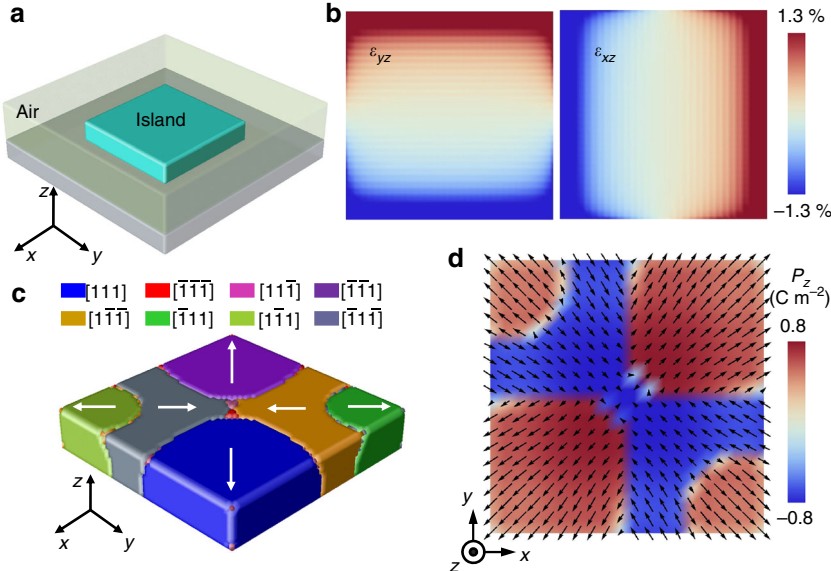

**Fig. 4** Phase field simulation of a radial quadrant domain structure. **a** Schematic for the system setting of phase field simulation. **b** Distributions of vertical shear strains $\varepsilon_{yz}$ and $\varepsilon_{xz}$ at bottom interface. **c** Three-dimensional view of the stabilised quadrant domain texture. The white arrows indicate the IP polarisation directions. **d** IP ferroelectric polarisation map on the top surface. The contrast represents the OOP polarisation

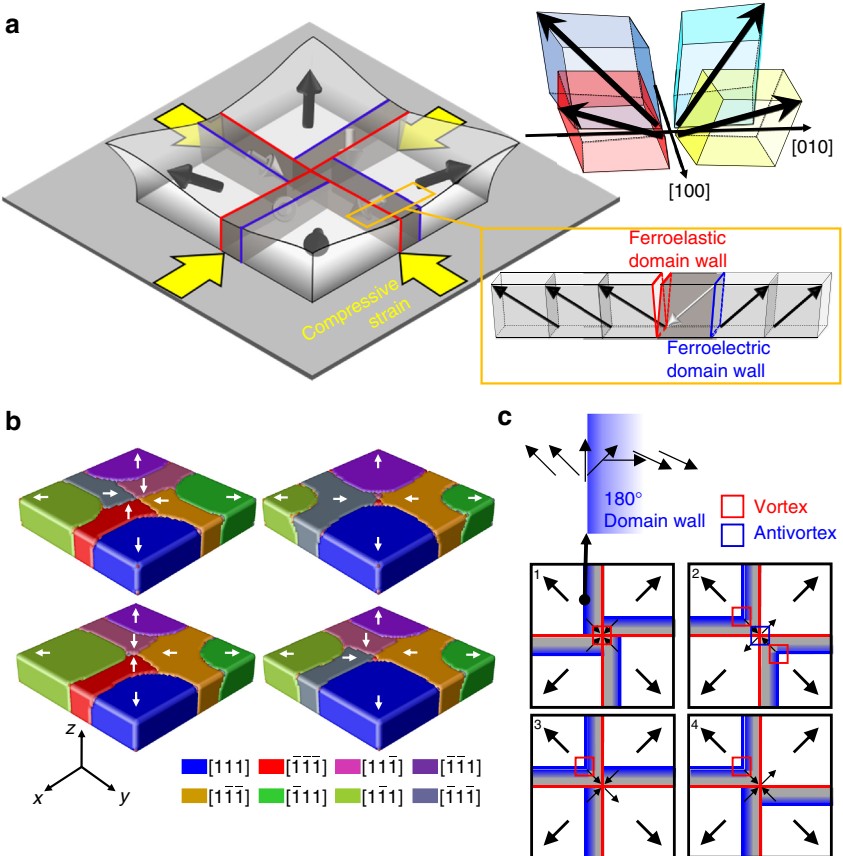

**Fig. 5** Radial quadrant domain structures in epitaxial BFO nanoplates subject to strain relaxation. **a** Schematic of a BFO rectangular nanoplate and an exotic quadrant domain structure therein. Red and blue lines represent the ferroelastic and ferroelectric domain walls, respectively. **b** Four representative buffer domain configurations stabilised by phase field simulation. **c** Schematics of the four types of domain configurations observed in the as-grown state. When one configuration can be superimposed on another configuration by symmetry operations of the point group 4 m, i.e., fourfold rotations, horizontal or vertical mirrors, their composite operations, both configurations are classified into the same type. Black arrows represent IP ferroelectric polarisations. The 180° ferroelectric domain walls are expressed by blue lines that form a semi-transparent gradient to represent the domain wall chirality (e.g., a clockwise Néel wall, as illustrated on the top). The expected vortex and antivortex points are marked on the schematics

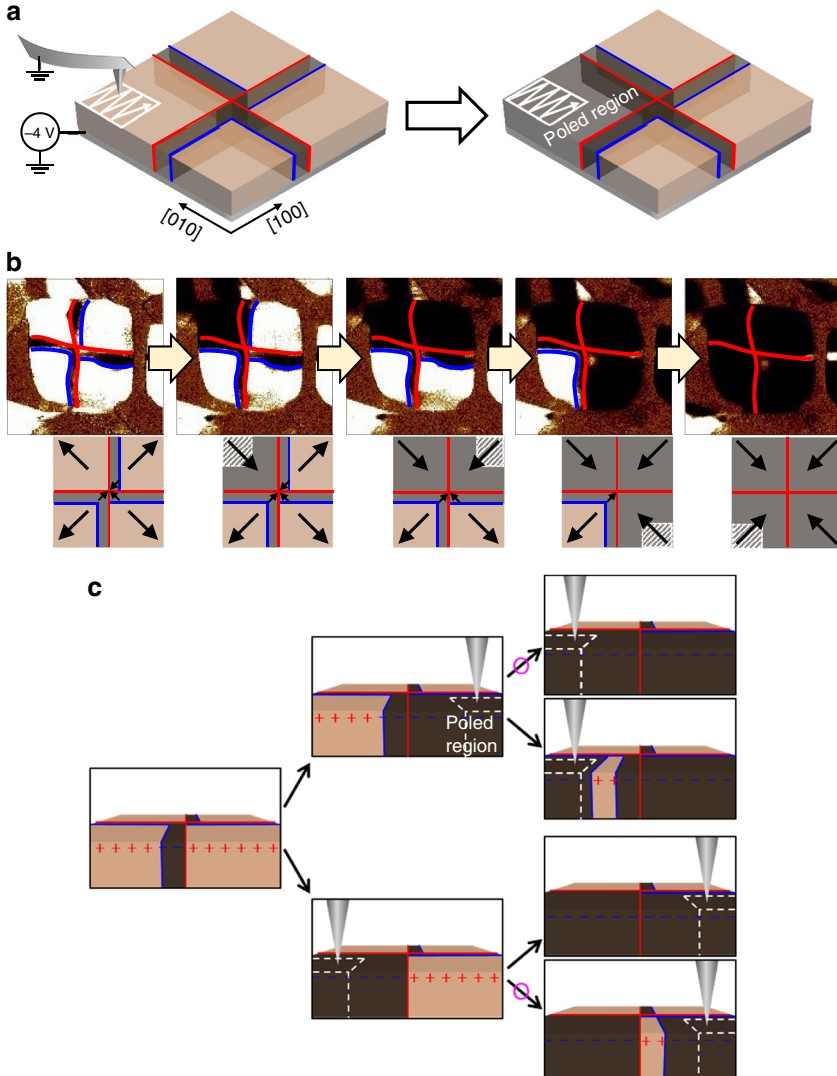

**Fig. 6** Non-local domain switching to control quadrant and buffer domains. **a** Schematic of a non-local switching process. **b** OOP PFM images showing domain switching one by one in a clockwise direction. The red and blue lines representing ferroelastic and ferroelectric domain walls, respectively. They were determined together with simultaneously measured IP PFM images (not shown here). **c** Buffer domain can be written or erased by control of the switching sequence. The pink circles indicate feasible routes verified by experiments

the buffer domain, so the switching forms a single ferroelectric domain in the corresponding quadrant without creating a buffer domain. As for the second domain switching, bound charges of the same polarity are located at the surface, so the existence of a buffer domain helps to reduce such unstable depolarisation energy.

**Configuration of the topological winding number**. We demonstrate that the system's total winding number can be configured artificially by electric fields. As the net winding number of topological defects is invariable for any continuous deformations of the order parameter, we should introduce a catastrophic transformation such as 180° polarisation switching of quadrant domain(s) to modify the total winding number. The total winding number of a BFO nanoplate relies on the relative arrangement of the quadrant and buffer domains. Provided that the polarisations of two neighbouring quadrant domains point outward (Supplementary Fig. 3), the partial winding number calculations along a line segment from a point in the first quadrant domain to another point in the second quadrant

domain are equally 1/4 (i.e. the angle of polarisation is finally increased by 90°) irrespective of the presence of a buffer domain. In the as-grown state, all quadrant domains point outward, so the total winding number is + 1 without regard to different buffer domain formations. On the contrary, if one points outward and the other points inward, the presence of a buffer domain between them increases the winding number by + 1 compared with the absent case. Using this rule, we inspected all possible domain configurations and found total winding numbers ranging from − 1 to 3.

We list each of the domain configurations eligible for the BFO nanoplate and classify them into five levels of topological invariants from a single antivortex to three vortices (Supplementary Fig. 4). Several representative configurations were chosen with four more total winding numbers in addition to the + 1 winding number of the as-grown states among the 1296 possible textures; these configurations are experimentally demonstrated in Fig. 7. The selective domain switching technique without invoking unintended pair creations was essential to stabilise the quasi-ground state. Remarkably, we found that electrical poling on the corner region (less than a quarter of a single quadrant

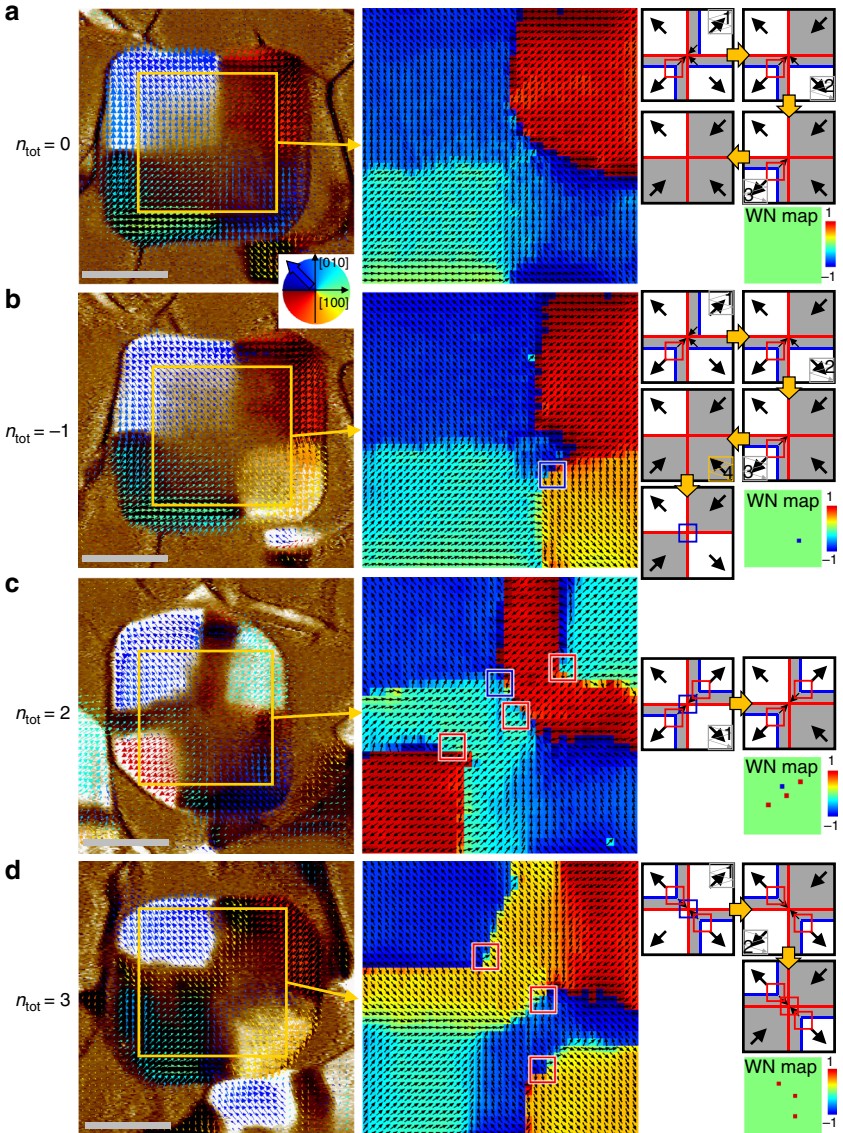

**Fig. 7** Reconfiguration of the total winding number of BFO nanoplates by selectively switching quadrant ferroelectric domains. Electrically controlled domain configurations show various total winding numbers, such as **a** $n_{tot} = 0$, **b** $n_{tot} = -1$, **c** $n_{tot} = 2$, and **d** $n_{tot} = 3$. Left: Vector maps of IP piezoresponse overlaid on the corresponding OOP PFM contrast; (middle) IP piezoresponse angle maps extracted from the yellow boxes; (right) Simplified schematics describing the domain switching sequence with expected vortex (red box) and/or antivortex (blue box) points; winding number maps were determined experimentally. The dark grey background regions have downward polarisations in contrast to the as-grown white regions with upward polarisations as a result of non-local ferroelectric switching of selected quadrant domains. It is noteworthy that the case of $n_{tot} = 2$ includes the creation of an unintended vortex–antivortex pair, but the measured total winding number is the same as the expected one that contains the two topologically protected vortices. Scale bars represent 200 nm

area) resulted in the reversal of the corresponding quadrant domain while inhibiting the production of a new domain wall. The non-local domain reversal provides a tremendous advantage by switching not only selective quadrant domains but also writing or erasing buffer domains deterministically by managing the switching sequence (see the schematics in the right-hand side column of Fig. 7), enabling us to realise all the possible configurations in principle.

We verify that interior modifications are unrelated to the total winding number. Figure 8 exhibits an excited state with vortex-antivortex pair creations. When all polarisations point inward without buffer domains, the central region becomes highly frustrated with competing local textures. Small electric perturbations easily induce new small domains via unintended pair creations. In the winding number map, many vortex–antivortex

pairs are clearly identified near the conjugate small domains, but the total winding number remains + 1 because of the topology.

**Evaluation of strain relaxation.** These exotic domain textures and switching behaviour are attributed to the radial strain relaxation from the residual compressive strain (as much as − 0.6% at the interface with edge dislocations; see Supplementary Fig. 5). Although the three-dimensional strain relaxation is complicated with the delicate features involved in spontaneous rhombic deformation, we obtained experimental hints regarding strain relaxation with the use of various diffraction techniques. The diffusive feature of the asymmetric peaks observed in X-ray reciprocal space maps (RSMs) is a clear signature of gradual strain relaxation (Supplementary Fig. 6). The broadening of the

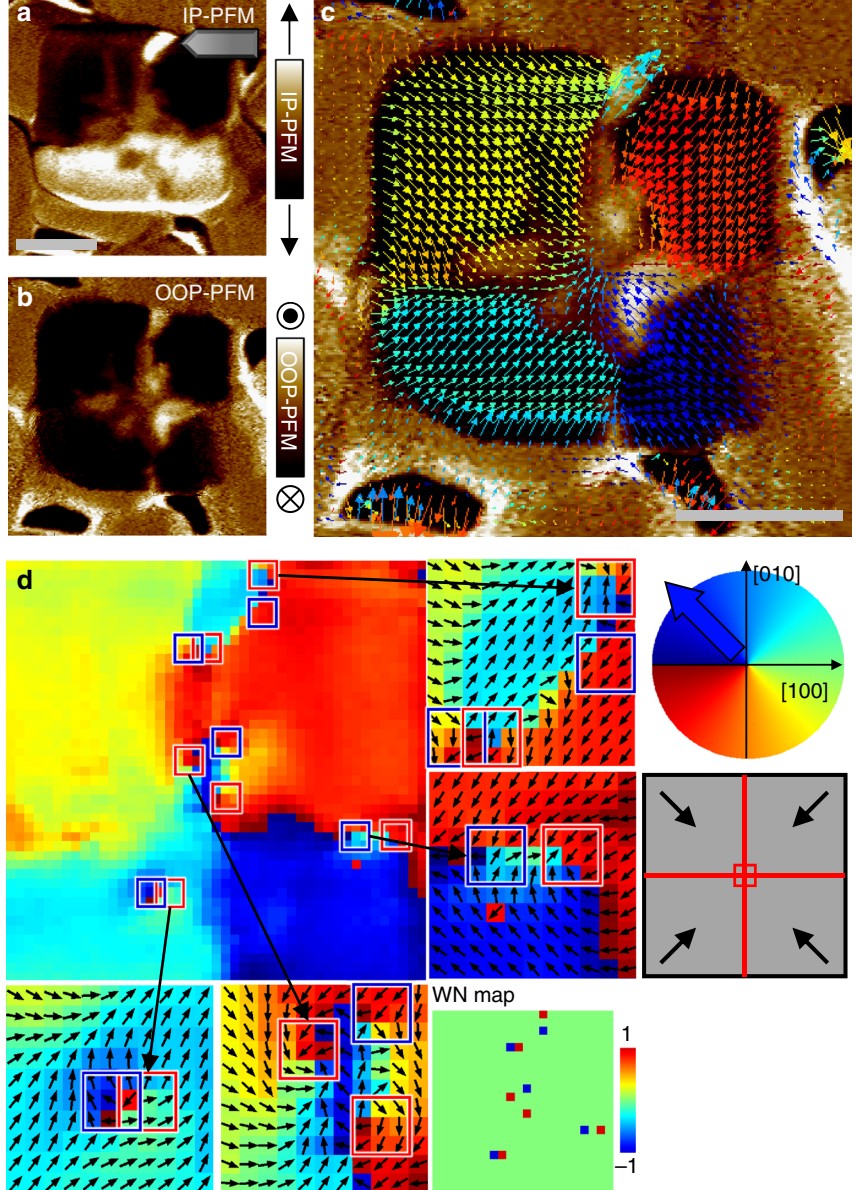

**Fig. 8** An excited state due to the creation of multiple vortex and antivortex pairs. **a**, **b** IP and OOP PFM images measured after electrical poling on the entire top surface of the plate with the type 2 state (the same plate in Fig. 3) by a *dc* biased tip at +3 V. The OOP polarisations in most areas were switched to downward polarisations (detected as dark contrast in the OOP PFM image) and flipped to inward IP polarisations. The same polarity of the polarisation merging at the central region produced a complex domain structure of relatively weak polarisation-up piezoresponses to partially avoid a strongly charged domain wall and reduce electrostatic energy. **c** IP piezoresponse vector map overlaid on OOP PFM contrast. **d** The corresponding colour map of IP piezoresponse directional angle. Vortex–antivortex pair generations are observed near polarisation-up regions. The central region contains one more vortex inevitable for this structural geometry, as explained in the schematic, thereby leading to a total winding number of 1. The electric frustration at the central region leads to the generation of various competing states and the strong electric perturbation during the whole-area poling offers more chances for excited states with multiple pairs of the particle and its anti-particle in the ferroelectric nanostructures subject to a strain gradient. Scale bars represent 200 nm

diffraction peak was also analysed with a Williamson-Hall plot and the inhomogeneous strains were quantitatively determined to be 0.60% and 0.46% for the OOP and IP lattice parameters, respectively (Supplementary Fig. 7a). Furthermore, the grazing incidence geometry enabled direct investigation of the dependence of the IP lattice parameter on the x-ray penetration depth (Supplementary Fig. 7b and c). These structural characterisations complementarily suggest that the compressive strain is almost relaxed within the 60 nm-thick nanoplates, resulting in a strain gradient whose order of magnitude is $10^5\,m^{-1}$. In addition, we

note that the strain gradient has a crucial role in the upward self-polarisation in the quadrant domains via the flexoelectric effect[36], because normal BFO films on the $Pr_{0.5}Ca_{0.5}MnO_3$ (PCMO) bottom electrode have been downward.

## Discussion

Our demonstration by using the unique visualisation approach and the winding number analysis not only offers a useful concept for multi-level topological defect memory, but also provides an avenue into strain-gradient-mediated clamping of topological

ferroelectric textures and non-local switching among symmetry-protected quantised states. The findings of this study also offer useful insights into electric pair creation, electric frustration and programmable charged domain walls. Of obvious future interest is an examination of the dynamic motion of the topological vortices along the chiral domain walls for energy efficient information technology: electric topological defectronics.

## Methods

**Growth of composite thin films.** BFO-CoFe$_2$O$_4$ (BFO-CFO) composite thin films were synthesised by pulsed laser deposition using a single target composed of Bi$_{1.1}$FeO$_3$ (65 atomic %) and CFO (35 atomic %). The self-assembled nanoplate structures of BFO-CFO were grown on (001) LaAlO$_3$ (LAO) substrates with a bottom electrode that comprised a ~ 4 nm-thick PCMO layer. Growths were made at 630 °C in an oxygen environment of 100 mTorr. A KrF excimer pulsed laser ($\lambda$ = 248 nm) was used at a frequency of 7 Hz to create a laser fluence of 0.8 J cm$^{-2}$ on the target surface. After the growths were completed, the samples were cooled to room temperature at a rate of 10 °C min$^{-1}$ in an oxygen environment of 500 Torr to minimise the current leakage of BFO.

**Angle-resolved PFM technique.** The surface topography and ferroelectric domains of BFO nanoplates were investigated with a scanning probe microscope (Bruker MultiMode V equipped with a Nanoscope controller V). PFM measurements were performed at a scanning rate of 3 μm s$^{-1}$ using Pt-coated Si conductive tips (MikroMasch, NSC35) applying an *ac* driving voltage of 2 $V_{pp}$ at a frequency 10 kHz in ambient conditions. In this study, all PFM images plot the real part piezoresponse signal, i.e., amplitude × cos(phase). In the OOP PFM images, the bright (dark) contrast represents upward (downward) polarisation. In the IP PFM images, the bright (dark) contrast indicates the IP piezoresponse vector pointing to [010] ([0–10]) when a cantilever is oriented towards [−100]. IP PFM exploits the torsional motion of the cantilever to sense the IP oscillations of the sample underneath. The IP PFM signal depends on the sample orientation with respect to the cantilever (i.e., the IP PFM signal is proportional to the projected component of the IP piezoresponse vector on the axis perpendicular to the cantilever). It was necessary to align these PFM images to correct pixel misalignment from an asymmetric tip shape and/or the tip-drift issue in nanoscale measurements. Several specific positions, including the domain walls and the corners of the nanoplates, were selected as reference points to determine the coordinate conversion matrices among the images. After this alignment, each position has tip-orientation-dependent IP PFM signals. Although a data pixel has a physical size of 3 × 3 nm, the signals in a single position for the vector maps are prepared by averaging over a 15 × 15 nm area to improve statics, considering our instrument resolution and sensitivity, and thus pair creations with a separation of less than the average distance are hardly detected. We also used finer 9 × 9 nm averaging for the angle maps to calculate the local winding number with 3-by-3 (27 × 27 nm) loops.

**Winding number calculation.** The vortex—a spatially confined object—has a core region with a discontinuous order parameter. Despite the singularity, the existence of a vortex affects the far-field region, where the order parameter changes slowly in space. Therefore, the presence of a vortex can be determined by measuring the variance of the order parameter such as the orientation of ferroelectric polarisation ($\theta$) on any closed contour that encloses the vortex core. Mathematically, the winding number $n$ in two-dimensional space is defined by the contour integral of an orientation change $\Delta\theta$ of two neighbouring IP ferroelectric polarisations along a given loop divided by $2\pi$. The winding number gives information regarding the net number of vortices and anti-vortices enclosed by the loop. For example, if a large closed loop encloses a pair of vortex and antivortex points with opposite winding numbers in a uniform far-field configuration of polarisation, the total winding number for the entire enclosed area is 0. We performed such winding number calculation as follows:

$$n = \frac{1}{2\pi} \oint_C \nabla\theta \cdot d\mathbf{r} \qquad (1)$$

where $\theta$ is measured anticlockwise from the [100] direction. We assume that $\theta$ is continuous everywhere, except for vortex or antivortex points, and we used a condition of $\Delta\theta < |180°|$ to determine the angle rotation direction. We tiled 3 × 3-pixel square loops on angle maps that shared the boundaries, to calculate the local winding number at each loop. A pixel of the angle maps includes an average IP PFM direction of a 9 × 9 nm area. Such a small loop has the advantage of measuring the precise position of each vortex or antivortex core.

**Phase field simulation.** In the phase field simulations, we introduce both polarisation, $P_i (i = 1–3)$, and oxygen octahedral tilt order parameters, $\theta_i (i = 1–3)$, to describe the domain structures in BFO. The total free energy density includes the contributions from the Landau bulk free energy, gradient energy, elastic energy and electrostatic energy:

$$F = \int_V \left[ \alpha_{ij} P_i P_j + \alpha_{ijkl} P_i P_j P_k P_l + \beta_{ij} \theta_i \theta_j + \beta_{ijkl} \theta_i \theta_j \theta_k \theta_l \right.$$
$$+ t_{ijkl} P_i P_j \theta_k \theta_l + \frac{1}{2} g_{ijkl} \frac{\partial P_i}{\partial x_j} \frac{\partial P_k}{\partial x_l} + \frac{1}{2} \kappa_{ijkl} \frac{\partial \theta_i}{\partial x_j} \frac{\partial \theta_k}{\partial x_l}$$
$$\left. + \frac{1}{2} c_{ijkl} \left( \varepsilon_{ij} - \varepsilon_{ij}^0 \right)\left( \varepsilon_{kl} - \varepsilon_{kl}^0 \right) - E_i P_i - \frac{1}{2} \varepsilon_0 \kappa_b E_i E_i \right] dV \qquad (2)$$

where $\alpha_{ij}$, $\alpha_{ijkl}$, $\beta_{ij}$, $\beta_{ijkl}$ and $t_{ijkl}$ are the coefficients of the Landau polynomial under stress-free boundary conditions, $g_{ijkl}$ and $\kappa_{ijkl}$ are the gradient energy coefficients, $x_i$ is the spatial coordinate, $c_{ijkl}$ is the elastic stiffness tensor, $\varepsilon_{ij}$ and $\varepsilon_{kl}^0$ are the total strain and eigenstrain, respectively, $E_i$ is the electric field, $\varepsilon_0$ is the permittivity of free space, and $\kappa_b$ is the background dielectric constant. The eigenstrain is related to the order parameters through $\varepsilon_{ij}^0 = h_{ijkl} P_k P_l + \lambda_{ijkl} \theta_k \theta_l + \varepsilon_{ij}^{\text{lattice}}$, where $\lambda_{ijkl}$ and $h_{ijkl}$ are coupling coefficients, and $\varepsilon_{ij}^{\text{lattice}}$ is eigenstrain caused by lattice parameter mismatch between BFO and the substrate. The current simulations use the parameters from Ref. [37], and a more comprehensive thermodynamic potential for BFO is available at Ref. [38].

To describe the mechanical boundary conditions of BFO nanoplates, the system consists of three types of materials, i.e., BFO, air and substrate. BFO possesses nonzero polarisation and the polarisation in the air and substrate is zero. The elastic stiffness of the air is zero and we assume that the elastic stiffness of the substrate is the same as BFO. Temporal evolution of the order parameter is described by the time-dependent Ginzburg–Landau equation, $\partial P_i/\partial t = -L_P(\delta F/\delta P_i)$ and $\partial \theta_i/\partial t = -L_\theta(\delta F/\delta \theta_i)$, which is solved numerically using the semi-implicit Fourier spectral method[39]. Periodic boundary conditions are applied along three directions, and a spectral iterative perturbation method is used to solve the mechanical and electrostatic equilibrium conditions[40]. Taking the lattice parameter of the substrate as the reference, $\varepsilon_{ij}^{\text{lattice}}$ can be calculated as $\varepsilon_{ij}^{\text{lattice}} = \frac{0.3965-0.3821}{0.3821} = 3.8\%$. To consider the effect of the depolarisation field along the out-of-plane direction, we calculate the average polarisation $\overline{P}_3 = \frac{\sum_{i=1}^{n} P_3}{n}$, and the depolarisation electric field $E_3 = -\frac{\overline{P}_3}{\varepsilon_b \varepsilon_0} + E_{\text{ex}}$, where $E_{\text{ex}}$ is the extra electric field caused by other factors such as the flexoelectric effect and its magnitude is tuned to obtain the domain structures similar to experiments.

The whole system grid is 128Δx × 128Δx × 60Δx with Δx = 0.38 nm, with 128Δx × 128Δx × 24Δx for the substrate and 64Δx × 64Δx × 12Δx for the BFO island. In the calculation of strain distributions, the order parameter $P_i$ and $\theta_i$ are maintained at zero. Thus, the distribution of strain is caused by the relaxation of the nanoplate rather than by the BFO domain structures.

**High-angle annular dark-field scanning TEM.** To gain more insight into the depth profile of the BFO nanoplate, we performed TEM for a cross-sectional view of a nano-composite thin film. As shown in Supplementary Fig. 5a, a low-magnification dark-field TEM image clearly revealed that a nanoplate with a lateral size of ~ 300 nm has a thickness of approximately 60 nm and a very thin PCMO layer that uniformly covers the substrate as a bottom electrode. CFO clusters were also seen around the nanoplate with relatively weak contrast in this Z-contrast image. To determine the epitaxial relationships among the nanoplate, the PCMO, and the substrate, we obtained high-angle annular dark-field (HAADF) images at the interfacial region of the central nanoplate (Supplementary Fig. 5b). The 4 nm-thick PCMO conducting layer was coherently deposited on LAO, that is, the IP lattice parameter of the PCMO was exactly matched with the substrate. However, the BFO nanoplate turned out to be partially relaxed at the initial stage, which created edge dislocations at the interface with the PCMO (red arrows in figure). The edge dislocations appear regularly with an interval of ~ 10 nm (~ 25 unit cells) along the interface at the measured central region. Accordingly, the IP lattice parameter of BFO at the central region is found to be relaxed to 3.94 Å, because the lattice expands by 1/25 from the lattice parameter (3.789 Å) of LAO. The origin of the regular edge dislocations that emerging at an interval of ~ 10 nm is inferred from a large lattice mismatch between the BFO and LAO substrate (~ 4.4% compressive strain compared with the lattice parameter of bulk BFO). Although a significant fraction of the strain is relaxed near the substrate, a compressive strain of ~ 0.6% remains and relaxes gradually across the BFO from the bottom to the top of the nanoplate. Furthermore, atomic-scale HAADF scanning TEM (STEM) images taken from the left, middle and right areas of the plate allowed us to check the local ferroelectric polarisation (Supplementary Fig. 5c-e), which could be identified by a relative Fe ion displacement with respect to the Bi cage. As the Fe ion of BFO drags the oxygen anions, which were unseen in our measurements, the position of the Fe ion enabled us to presume the negative centre of a unit cell. The observed Fe ion shifts led us to the conclusion that the local polarisations on the left and right areas were upward and outward. Meanwhile, the Fe off-centring at the middle area was downward, and the magnitude was less than those of the outer areas. These observations showed good agreement with the radial quadrant domain structure verified by the angle-resolved PFM. For the cross-sectional observation on the BFO nanoplate, the specimens were prepared by a dual-beam focused ion beam system (JIB-4601F, JEOL, Japan). To protect the BFO plates and CFO films, an amorphous carbon layer was deposited on the top surface before ion beam milling. A Ga$^+$ ion beam with an acceleration voltage of 30 kV was used to fabricate

the thin TEM lamella. To minimise the surface damages induced by Ga$^+$ ion beam milling, the sample was further milled with an Ar$^+$ ion beam (PIPS II, Gatan, USA) with an acceleration voltage of 0.8 kV for 15 min. Z-contrast HAADF STEM images were taken with a scanning transmission electron microscope (JEM-2100F, JEOL) at 200 kV with a spherical aberration corrector (CEOS GmbH, Germany). The optimum size of the electron probe was approximately 0.9 Å. The collection semi-angles of the HAADF detector were adjusted from 80 to 220 mrad to exploit the large-angle elastic scattering of electrons for clear Z-sensitive images. The raw images obtained were processed with a Wiener filter with a local window to reduce background noise (HREM Research Inc., Japan).

**X-ray diffraction**. RSMs: X-ray diffraction measurements were carried out at beamline 3 A of the Pohang Light Source with a wavelength of 0.9428 Å. To investigate the lattice parameters in detail, we measured RSMs around the asymmetric crystallographic peaks of randomly distributed ~ 60 nm-thick nanoplates in a BFO-CFO composite film grown on an LAO substrate. As shown in Supplementary Fig. 6, two distinct BFO phases (R and T) with significantly different c-axis lattice parameters were detected on both (104) and (114) RSMs. By analysing the two RSMs, we determined the pseudo-cubic symmetry of each phase and lattice parameters. The peaks of our main interest R-BFO have diffusive shapes with their tails pointed toward smaller IP reciprocal positions, indicating the existence of significant strain relaxation within the nanoplates. In particular, we observed a more prominent diffusive feature in the (114) RSM than the (104) RSM and inferred that a more well-defined strain gradient appears along the [HHL] crystallographic axis.

Williamson–Hall plots: The inhomogeneous strain distribution within nanoplates was measured quantitatively in the context of the Williamson–Hall plot[41,42]. First, we performed longitudinal $\theta$–$2\theta$ scans for OOP (00L) peaks up to the fifth order and collected diffraction patterns of the IP (H00) peaks using grazing incidence geometry. We then examined the evolution of the peak broadenings depending on the order (H or L) of the peaks. The broadness of the diffraction peaks can be influenced by the size-broadening ($\beta_L$) effect (e.g., film thickness) and the strain-broadening ($\beta_e$) effect (e.g., a strain gradient along the longitudinal direction). These two effects can be distinguished by their different dependences on the Bragg angle $\theta$, i.e. $\beta_L \sim \frac{0.9\lambda}{t \cos\theta}$ and $\beta_e \sim 4\varepsilon_I \tan\theta$, where $\lambda$ is the wavelength of an incident X-ray beam, $t$ stands for the film thickness and $\varepsilon_I$ represents the SD of the strain distribution known as inhomogeneous strain. By combining the two broadenings in a linear regime, the intrinsic line broadening ($\beta$) of diffraction peaks can be written as $\beta\cos\theta \sim 4\varepsilon_I \sin\theta + \frac{0.9\lambda}{t}$. Accordingly, the Williamson–Hall plot ($\beta\cos\theta$ vs. $4\sin\theta$) gives information regarding the inhomogeneous strain from the slope and the film thickness from the intercept. The values of $\varepsilon_I$ for the c-axis and a-axis lattice parameters were determined to be 0.0060 and 0.0046, respectively (Supplementary Fig. 7a). The linear fitting lines in the plot were constrained in such a way that they intersected at an identical point at $\theta = 0$. From the intercept, the film thickness is determined to be ~ 60 nm, which is consistent with the TEM cross-sectional image.

Grazing incidence XRD: A variation of the IP lattice parameter was directly measured as a function of the probing depth ($d_p$) by using the grazing incidence geometry (Supplementary Fig. 7b). The penetration depth of the X ray is nearly proportional to the incidence angle $\mu$, according to $d_p = \lambda \sin\mu / 4\pi\beta^*$, where the effective imaginary part of reflective index $\beta^*$ was set to be the BFO value ($4.3 \times 10^{-7}$)[43]. The depth profile shows that the lateral strain ($\varepsilon_{xx}$ and $\varepsilon_{yy}$) is rapidly relaxed near the surface (Supplementary Fig. 7c). To explain the observed strain depth profile, the strain $\varepsilon(z)$ was written as $\varepsilon(z) = -\varepsilon_1 + \varepsilon_0 e^{\alpha z}$ where $\varepsilon_1$, $\varepsilon_0$ and $\alpha$ are coefficients, and $z$ is the distance from the substrate. We fit the average strain model $\langle\varepsilon\rangle_{d_p} = \frac{1}{d_p}\int_0^{d_p}\varepsilon(t-z')\mathrm{d}z'$ to the experimental data, where $t$ denotes the film thickness (60 nm) and the variable of integration $z'$ measures from the top surface. The coefficients $\varepsilon_1$, $\varepsilon_0$ and $\alpha$ were determined to be 0.0043, $2.6 \times 10^{-9}$ and 0.24 nm$^{-1}$, respectively. The fitting was well matched with an $R^2$ value of 0.9874. We thus estimate the average strain gradient across the entire BFO nanoplate $\langle\frac{de}{dz}\rangle_t$ as $0.8 \times 10^5$ m$^{-1}$, which agrees well with the inhomogeneous strain.

**Data availability**. The authors declare that the data supporting the findings of this study are available within the paper and its Supplementary Information files. Additional data are available from the corresponding author upon reasonable request.

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

## Acknowledgements

This work was supported by the National Research Foundation (NRF) Grant funded by the Korean Government via the Creative Research Initiative Center for Lattice Defectronics (2017R1A3B1023686) and the Center for Quantum Coherence in Condensed Matter (2016R1A5A1008184). X-ray studies were carried out at the Pohang Accelerator Laboratory. This work was also supported by the Global Frontier R&D Program on Center for Hybrid Interface Materials (HIM) funded by the Ministry of Science, ICT & Future Planning (2013M3A6B1078872). The work at Penn State was supported by the U.S. Department of Energy, Office of Basic Energy Sciences, Division of Materials Sciences and Engineering under Award FG02-07ER46417 (F.X. and L.-Q.C.) and by the Penn State MRSEC, Center for Nanoscale Science, under the award NSF DMR-1420620 (F.X.).

## Author contributions

K.-E.K. and C.-H.Y. conceived and designed the project. K.-E.K. and S.J. prepared samples. K.-E.K., S.J. and K.C. measured and analysed the angle-resolved PFM with the winding number calculation. J.H.L and T.Y.K. performed structural analysis by X-ray diffraction. G.-Y.K. and S.-Y.C. carried out STEM. F.X. and L.-Q.C. performed phase field simulation. K.-E.K., R.R. and C.-H.Y. led manuscript preparation with contributions from all authors.

## Additional information

**Competing interests:** The authors declare no competing financial interests.

