## [Peer Review File · Nature Communications]

PEER REVIEW FILES

Reviewers' comments:

Reviewer #1 (Remarks to the Author):

1. The paper reports observations of a strained bismuth-ferrite nanoplate; it is shown that the nanoplate adopts a pattern of domains including some vortex-like features. The authors show that this structure can be changed by electric field; the winding number of the polarization field is conserved during small perturbations of the structure, but can be changed by selective ferroelectric switching of parts of the nanoplate.
2. The main claim to novelty appears to be the stabilization, by misfit strain on a substrate, of a ferroelectric vortex structure. This claim appears valid as the authors demonstrate observations of the structure by several techniques. A second significant claim is the ability to manipulate the structure using electric fields. Again, this appears valid as the authors show measurements of the structure after selective switching of domains.
3. The findings certainly will be of interest to the community and the wider field. There are potential applications in memory devices and other areas of technology.
4. Noting the above significant claims to novelty, the paper could be published. There are also, however, some aspects that could be called into question. The analysis may be correct but is not always convincing. Comments on these aspects follow.
5. The assumption that the piezoresponse vector matches the ferroelectric polarization may not be correct for BFO. Consequently the various PFM images may not be showing exactly what the authors take them to be showing.
6. There is a disappointing mismatch between the model and the experimental data in the sense that the model does not appear to show at all the buffer regions noted in the experimental observations. These buffer regions appear essential to the formation of vortex-like features in the experiment, so the model does not show the key feature of interest. Given this, it is unclear what the model adds to the exposition.
7. The language is unclear in places, for example at line 168, "This axial switching behavior provides the mathematical logic to digitally manipulate the number of vortices..." It is unclear what mathematical logic has to do with this. Similarly at line 59, "The other side connection

between the curved lattice space and the non-trivial texture is also robust, because the original topology is mathematically concerned with space properties". These sentences and others like them will be unintelligible to most readers.

8. There have been numerous publications on vortex-type structures, and it is not clear that this work would greatly influence thinking in the field. However, it is of interest.

9. On balance, the paper could be published, but revisions would ideally be needed to address points 5, 6 and 7 above, including an overhaul of the language for improved clarity.

Reviewer #2 (Remarks to the Author):

The manuscript by Yang et al. reports an exciting experimental work on the complex vortex textures in strained BiFeO₃ nano-plates, which can be reconfigured by applying a tip bias for domain switching. Vortices and Skyrmion states in ferroelectrics have been a hot topic in recent years. Using the angle-resolved piezoresponse force microscopy, local winding number analysis, phase-field simulation, and several traditional characterization tools, the authors demonstrate the existence of vortices/anti-vortices, the pair creation and annihilation, and the manipulation of these exotic features. I agree with them that all these are major scientific advances and should be properly documented. My main objection to the publication in its present form lies in the poor readability. In my opinion, narratives like this paper represent an unhealthy trend in scientific literature and should be corrected soon.

The paper reads as if it tries to squeeze very rich contents into a few pages -- I suspect that it is transferred from other Nature journals where the page limit is very strict. Here are some examples.

1) The introduction does not allow the audience to transition from very abstract concepts of vortices and skyrmions to real material systems. The needed information is distributed to Methods (too long even for Nature Communications' 3000-word limit) and Supplementary Figures 3, 8, 9, and 10. While some of them are indeed technical details, I believe that proper descriptions of the samples and large area PFM data (as in Fig S3) are crucial to prepare the readers on the main topic.

2) The figure captions are also extremely lengthy, which appears to make an effort to circumvent the length limit of the main text. In fact, I think the caption of Figure 2 may have gone over the 350-word limit. The caption should be there to describe the figure only, leaving discussions and analysis to the text.

3) The construction of the in-plane piezo-response vector (SI, Fig. S1) is a very important experimental detail to justify the observation. I strongly suggest that this be moved to the main text.

4) The manipulation of vortices by tip writing (Fig. S7) is central to their claims. It should somehow appear in the text before Fig. 4.

In all, I think the authors should completely rewrite the manuscript as a long scholarly paper that transitions naturally between different sections and contains the necessary details in the flow. It may still fit to the page limit of Nature Communications. If not, I'd suggest that they go to PRB as an alternative.

Reviewer #3 (Remarks to the Author):

This paper experimentally studies the domain patterns in nanoplatelets of BFO in the rhombohedral phase. The domains are imaged using piezo force microscopy. A variety of quadrant domain patterns with different topological structures are obtained. It is further shown how to switch between patterns using external electric fields. I find these results very interesting and may lead to possible applications in memory devices by utilizing domain switching between large number of quadrant domain patterns. From a fundamental science point of view, I found this to be the first study reporting on domain patterns in rhombohedral ferroelectrics in nanoscale. Topological such as vortices are usually studied in context of tetragonal ferroelectrics, which are simpler than the rhombohedral case as they have fewer polar variants. The complexity of the nanoscale rhombohedral patterns is nicely shown in this paper. I strongly recommend to publish this paper in nature communications. However, I would like the authors to address the following questions.

1. Role of strain gradients: What is the role played by the strain gradient in stabilizing the quadrant structures. Such structures could simply arise from an interplay between elastic (electrostrictive) and depolarization fields. Since the authors are emphasizing on strain gradients, the question that arises is whether effects like flexoelectricity are playing a role in domain formation.

2. How are the five configurations shown in Fig 3 realized ? What was the initial pattern. It would be helpful to describe the switching sequence used to obtain these structures, at least for one or two of the cases shown.

3. Phase Field Simulation: My main criticism of this work is about the phase field simulation

performed in the paper. It is not at all clear how the simulations are done. First of all, Landau theory used does not appear to be correct. There is no coupling term between the polarizations, without which it is not possible to stabilize the rhombohedral state. The full Landau theory for BFO has already been derived and used in phase field simulations (J. X Zhang et. al Journal of Applied Physics 103, 094111 (2008), W.L. Cheah et al. Acta Materialia 100 (2015) 323–332). Why authors still used the simplified model with normalized parameters ?

How are the depolarization fields and long-range elasticity taken into account ? What are the boundary conditions which are used in simulations where the authors predict the distribution of the strains. Without getting a clear picture of the simulation methodology, it is difficult to assess how the simulation results support the experimental results.

Reviewers' comments:

Reviewer #1 (Remarks to the Author):

Review#1-1 :

1. The paper reports observations of a strained bismuth-ferrite nanoplate; it is shown that the nanoplate adopts a pattern of domains including some vortex-like features. The authors show that this structure can be changed by electric field; the winding number of the polarization field is conserved during small perturbations of the structure, but can be changed by selective ferroelectric switching of parts of the nanoplate.

2. The main claim to novelty appears to be the stabilization, by misfit strain on a substrate, of a ferroelectric vortex structure. This claim appears valid as the authors demonstrate observations of the structure by several techniques. A second significant claim is the ability to manipulate the structure using electric fields. Again, this appears valid as the authors show measurements of the structure after selective switching of domains.

3. The findings certainly will be of interest to the community and the wider field. There are potential applications in memory devices and other areas of technology.

4. Noting the above significant claims to novelty, the paper could be published. There are also, however, some aspects that could be called into question. The analysis may be correct but is not always convincing. Comments on these aspects follow.

Response :

We would like to thank the referees for their constructive comments on our paper, which have helped us improve the manuscript.

Review#1-2 :

5. The assumption that the piezoresponse vector matches the ferroelectric polarization may not be correct for BFO. Consequently the various PFM images may not be showing exactly what the authors take them to be showing.

Response :

We understand the referee's concern. The piezoresponse vector indicates the movement of the tip contact point due to converse piezoelectric effect on the underlying material in response to an inhomogeneous electric field generated by a biased tip. Although the piezoresponse vector tends to correlate with ferroelectric polarization, the relation is not strict especially in the system where multiple domains are existent and can be influenced by interfacial geometry and orientation-dependent materials parameters. Although the PFM images can provide a useful insight into ferroelectric domain structures, we should be always cautiously aware of the limitation.

A difficulty in interpreting PFM signals arises from the fact that a material exhibits different piezoelectric coefficients, which, depending on the direction of the electric field, lead to a

longitudinal, a transversal or a shear deformation of the sample. A clear assignment of the PFM signals to torsion, buckling and deflection of the cantilever is also necessary. [E. Soergel, *J. Phys. D: Appl. Phys.* 44, 464003 (2011), S. V. Kalinin, *et al. Microsc. Microanal.* 12, 206 (2006)]. In particular, many papers have reported on lateral signals at domain boundaries. It has been reported that the topographical slope at the domain boundary caused by opposite deformations on antiparallel domains results in the torsion of the cantilever [see Fig. R1; D. A. Scrymgeour, *et al. Phys. Rev. B* 72, 024103 (2005), J. Wittborn, *et al. Appl. Phys. Lett.* 80, 1622 (2002), F. Johann, *et al. Appl. Phys. Lett.* 97, 102902 (2010), J. Guyonnet, *et al. Appl. Phys. Lett.* 95, 132902 (2009)]. Detailed theoretical analyses identified nonzero shear components at the 180° domain walls between up and down polarized domains through effective piezoresponse d_{35} due to symmetry breaking at the domain walls [A. N. Morozovska, *et al. Phys. Rev. B* 75, 174109 (2007)].

Although we cannot exclude the possibility of the unusual piezoresponse inherent especially at 180° domain walls, it is also true that the creation of vortex points cannot be explained by only *local* intrinsic/extrinsic effects at domain walls. Winding number calculation along any closed loop enclosing a single vortex point gives the same number (+1). The vortex point seems to be highly localized and determined by its local environment. In reality, topological vortices are a matter of global symmetry and their existence is protected from any intrinsic/extrinsic spatial and temporal fluctuations. Accordingly, we believe our topological argument is relatively safe than other PFM-based topics at domain walls. Furthermore, the domain structures with the emergence of the topological vortices can be supported by more rigorous phase field simulation (as will be addressed in the following question) as well as the topological analysis. These comprehensive efforts will help us to lead to more convincing arguments.

We thank the referee for raising this important issue. To relieve the concern of the referee, we leave a cautious note about the possible unusual PFM signal in the revised manuscript with citing one reference, as below.

“Although the PFM vector map provides a useful insight into ferroelectric domain structures under the assumption that piezoresponse vector has a linear correlation with electric polarisation, we should be cautiously aware of the limitation. For example, it has been reported that 180° domain walls can generate a lateral piezoresponse due to a topographical slope at the domain boundary caused by opposite deformations on the neighbouring up and down polarised domains³¹. We are not sure how largely the effect is involved in our case, and thus, it is desirable to interpret the detailed feature of the domain walls based on theoretical supports through topological analysis and phase field simulation.”

Figure R1: When positive voltage is applied to the tip, the cantilever torsions to the right in the slope model

Review#1-3 :

6. There is a disappointing mismatch between the model and the experimental data in the sense that the model does not appear to show at all the buffer regions noted in the experimental observations. These buffer regions appear essential to the formation of vortex-like features in the experiment, so the model does not show the key feature of interest. Given this, it is unclear what the model adds to the exposition.

Response :

We thank the referee for raising this important criticism. We didn't consider the depolarization field in the previous free energy model because of our limited computational power, even if the long-range interaction is most likely responsible for the emergence of the buffer domains. After receiving the question, we invited an expert group of phase field simulation to describe our experimental observations in a more rigorous way. In the new calculations, we used the Landau model with realistic coefficients for BFO (please see the Method part for the simulation details).

We confirm that the depolarization energy plays a critical role in creating the buffer domains, otherwise, all the regions have the same out-of-plane polarizations without the buffer domains. In addition, it is worth noting that the flexoelectric polarization competes with the depolarization field effect, thereby determining the areas of the buffer domain regions. We fully reproduce the possible as-grown domain structures (see Fig. R2 below). This updated phase field simulation results are the major achievement we made during this revision and they are reflected in Figs. 4 & 5 and Fig. S2 in the revised manuscript.

Figure R2: All possible domain configurations in the as-grown state. The white arrows indicate the electric polarization. The domain of the electric polarization toward the center of nanoplate is the buffer domain. We confirmed that the domain configurations of the four cases expected in the as-grown state are stabilized.

Review#1-4 :

7. The language is unclear in places, for example at line 168, “This axial switching behavior provides the mathematical logic to digitally manipulate the number of vortices...” It is unclear what mathematical logic has to do with this. Similarly at line 59, “The other side connection between the curved lattice space and the non-trivial texture is also robust, because the original topology is mathematically concerned with space properties”. These sentences and others like them will be unintelligible to most readers.

Response :

The two sentences are removed in the revised manuscript to avoid any confusion. We have rearranged the description sequence and rewritten many parts of the paper to improve readability and match the manuscript to the Article format.

Review#1-5 :

8. There have been numerous publications on vortex-type structures, and it is not clear that this work would greatly influence thinking in the field. However, it is of interest.

Response :

In this paper, we demonstrate that topologically non-trivial ferroelectric domain structures can emerge due to inhomogeneous strain distribution. This mechanical constraint is attributed to misfit strain relaxation under the anisotropic mechanical boundary conditions of the nanoplate. In aspect of methodology, the angle-resolved PFM in conjunction with the winding number analysis offers a powerful way into clarifying ferroelectric topological structures.

Review#1-6 :

9. On balance, the paper could be published, but revisions would ideally be needed to address points 5, 6 and 7 above, including an overhaul of the language for improved clarity.

Response :

We thank the referee for supporting our manuscript and giving us the constructive comments.

Reviewer #2 (Remarks to the Author):

Review#2-1 :

The manuscript by Yang et al. reports an exciting experimental work on the complex vortex textures in strained BiFeO₃ nano-plates, which can be reconfigured by applying a tip bias for domain switching. Vortices and Skyrmion states in ferroelectrics have been a hot topic in recent years. Using the angle-resolved piezoresponse force microscopy, local winding number analysis, phase-field simulation, and several traditional characterization tools, the authors demonstrate the existence of vortices/anti-vortices, the pair creation and annihilation, and the manipulation of these exotic features. I agree with them that all these are major scientific advances and should be properly documented. My main objection to the publication in its present form lies in the poor readability. In my opinion, narratives like this paper represent an unhealthy trend in scientific literature and should be corrected soon.

The paper reads as if it tries to squeeze very rich contents into a few pages -- I suspect that it is transferred from other Nature journals where the page limit is very strict. Here are some examples.

Response:

We would like to thank the referee for his/her constructive comments to help us significantly improve the manuscript. We follow the referee's considerate suggestions on rearranging the document and we are very glad that the revised manuscript in the Article format is more informative and smoothly connect the abstract topological concept to the real material system. By transferring considerable parts of Supplementary Figures and Methods section to the main paper and incorporating new schematics and descriptions to deliver the detail procedures, the paper becomes better organized, thereby the readability is significantly improved.

Review#2-2 :

1. The introduction does not allow the audience to transition from very abstract concepts of vortices and skyrmions to real material systems. The needed information is distributed to Methods (too long even for Nature Communications' 3000-word limit) and Supplementary Figures 3, 8, 9, and 10. While some of them are indeed technical details, I believe that proper descriptions of the samples and large area PFM data (as in Fig S3) are crucial to prepare the readers on the main topic.

Response :

As asked by the referee, we have moved the large area PFM data (Fig. S3 in the original manuscript) to the main Fig. 1 in the revised manuscript. We also provide an overall description of the sample in the main text related to the figure, so that the readers can understand the real system before jumping into the topic regarding topological properties.

Review#2-3 :

2. The figure captions are also extremely lengthy, which appears to make an effort to circumvent the length limit of the main text. In fact, I think the caption of Figure 2 may have gone over the 350-word limit. The caption should be there to describe the figure only, leaving

discussions and analysis to the text.

Response :

As mentioned by the referee, the caption of Fig. 2 in the original manuscript exceeded the word limit. So, we move the discussion/analysis part to the main text leaving only the text directly related to the corresponding figure in the caption.

Review#2-4 :

3. The construction of the in-plane piezo-response vector (SI, Fig. S1) is a very important experimental detail to justify the observation. I strongly suggest that this be moved to the main text.

Response :

As suggested by the referee, we move the Fig. S1 to the main Fig. 2 of the revised manuscript. The contents of the in-plane angle-resolved PFM in the Methods section are also transferred to the relevant main text for a detailed description of the experiment.

Review#2-5 :

4. The manipulation of vortices by tip writing (Fig. S7) is central to their claims. It should somehow appear in the text before Fig. 4.

Response :

Fig. S7 (in the original manuscript) introduced our experimental switching sequence of the quadrant domains to write and erase the buffer domain. We fully agree with the referee's opinion that the Fig. S7 plays an important role in our claim that the total winding number of BFO nanoplates can be configured by selectively switching quadrant ferroelectric domains, so we move Fig. S7 to the main Fig. 6 that is located before the demonstration (Fig. 7 in the revised manuscript). The part of non-local domain switching in the Methods section is also moved to the main text related to the figure.

Review#2-6 :

In all, I think the authors should completely rewrite the manuscript as a long scholarly paper that transitions naturally between different sections and contains the necessary details in the flow. It may still fit to the page limit of Nature Communications. If not, I'd suggest that they go to PRB as an alternative.

Response :

We appreciate the referee's comments on the sequence of contents for the smooth connection. By incorporating all the referee's suggestions, the revised manuscript becomes more informative in understanding the BFO nanoplate and switching details, and we believe it is now clearer and easier to understand than the previous version. Although many of the Supplementary Figures and Method parts are moved to the main paper, it does not exceed the page limit (5000 words & 10 display items) of Nature Communications.

Reviewer #3 (Remarks to the Author):

Review#3-1 :

This paper experimentally studies the domain patterns in nanoplatelets of BFO in the rhombohedral phase. The domains are imaged using piezo force microscopy. A variety of quadrant domain patterns with different topological structures are obtained. It is further shown how to switch between patterns using external electric fields. I find these results very interesting and may lead to possible applications in memory devices by utilizing domain switching between large number of quadrant domain patterns. From a fundamental science point of view, I found this to be the first study reporting on domain patterns in rhombohedral ferroelectrics in nanoscale. Topological such as vortices are usually studied in context of tetragonal ferroelectrics, which are simpler than the rhombohedral case as they have fewer polar variants. The complexity of the nanoscale rhombohedral patterns is nicely shown in this paper. I strongly recommend to publish this paper in nature communications. However, I would like the authors to address the following questions.

Response :

We sincerely appreciate the referee's positive evaluation on our manuscript. In the following, we addressed and incorporated the referee's questions and suggestions.

Review#3-2 :

1. Role of strain gradients: What is the role played by the strain gradient in stabilizing the quadrant structures. Such structures could simply arise from an interplay between elastic (electrostrictive) and depolarization fields. Since the authors are emphasizing on strain gradients, the question that arises is whether effects like flexoelectricity are playing a role in domain formation.

Response :

We are grateful for the referee' valuable comments. From the phase field simulation, the quadrant domains are caused by the inhomogeneous shear strain, and the inhomogeneous shear strain is caused by the relaxation along the thickness direction. Flexoelectricity affects the presence of the buffer domain. If there is no flexoelectric field, the entire two diagonal domains will have downward polarizations rather than the formation of buffer domains. Fig. S2a shows that the strain-gradient-induced flexoelectric effect is strongest at the four corners. Therefore, the polarization should point up, and the depolarization field can only flip down the polarization in the middle part of nanoplate. The competition between depolarization energy and flexoelectricity determines the position and size of the buffer domains.

Review#3-3 :

2. How are the five configurations shown in Fig 3 realized ? What was the initial pattern. It would be helpful to describe the switching sequence used to obtain these structures, at least for one or two of the cases shown.

Response :

This is a great idea to help readers understand our experiments. To address the switching sequence, we include new schematics on the right-hand side column of the Fig. 7 in the revised manuscript (Fig. 3 is moved to Fig. 7).

Review#3-4 :

3. Phase Field Simulation: My main criticism of this work is about the phase field simulation performed in the paper. It is not at all clear how the simulations are done. First of all, Landau theory used does not appear to be correct. There is no coupling term between the polarizations, without which it is not possible to stabilize the rhombohedral state. The full Landau theory for BFO has already been derived and used in phase field simulations (J. X Zhang et. al Journal of Applied Physics 103, 094111 (2008), W.L. Cheah et al. Acta Materialia 100 (2015) 323–332). Why authors still used the simplified model with normalized parameters ?

How are the depolarization fields and long-range elasticity taken into account ? What are the boundary conditions which are used in simulations where the authors predict the distribution of the strains. Without getting a clear picture of the simulation methodology, it is difficult to assess how the simulation results support the experimental results.

Response :

After receiving the questions, we performed new phase field simulations using more accurate Landau free energy with realistic parameters of BFO and also included the depolarization fields. The free energy equation, coefficients, depolarization field calculation, and boundary conditions we used are as follows.

1. The free energy model :

In the phase-field simulations, we introduce both polarization, $P_i (i=1-3)$, and oxygen octahedral tilt order parameters, $\theta_i (i=1-3)$, to describe the domain structures in BFO. The total free energy includes the bulk free energy, gradient energy, elastic energy, and electrostatic energy.

$$F_{total} = \int_V [f_{bulk}(p, \theta) + f_{grad}(p, \theta) + f_{elec}(p) + f_{elas}(p, \theta)] dV$$

Bulk free energy : $f_{bulk} = \alpha_{ij} P_i P_j + \alpha_{ijkl} P_i P_j P_k P_l + \beta_{ij} \theta_i \theta_j + \beta_{ijkl} \theta_i \theta_j \theta_k \theta_l + t_{ijkl} P_i P_j \theta_k \theta_l,$

Gradient energy : $f_{grad} = \frac{1}{2} g_{ijkl} P_{i,j} P_{k,l} + \frac{1}{2} \kappa_{ijkl} \theta_{i,j} \theta_{k,l},$

Elastic energy : $f_{elas} = \frac{1}{2} c_{ijkl} (\epsilon_{ij} - \epsilon_{ij}^0)(\epsilon_{kl} - \epsilon_{kl}^0), \quad \epsilon_{ij}^0 = h_{ijkl} P_k P_l + \lambda_{ijkl} \theta_k \theta_l + \epsilon_{ij}^{lattice}$

Electrostatic energy : $f_{elec} = -\frac{1}{2} E_i p_i - \frac{1}{2} \epsilon_0 \epsilon_r E_i E_i,$

where $\alpha_{ij}, \alpha_{ijkl}, \beta_{ij}, \beta_{ijkl}$, and t_{ijkl} are the coefficients of the Landau polynomial under stress-free boundary conditions, g_{ijkl} and κ_{ijkl} are the gradient energy coefficients, c_{ijkl} is the elastic stiffness tensor, ε_{ij} and ε_{kl}^0 are the total strain and eigenstrain which is the strain produced without external forces, respectively. λ_{ijkl} and h_{ijkl} are coupling coefficients, and $\varepsilon_{ij}^{\text{lattice}}$ is eigenstrain caused by lattice parameter mismatch between BFO and the substrate. E_i is the electric field, ε_0 is the permittivity of free space, and ε_r is the dielectric constant.

2. The coefficients used in the simulation :

All the coefficients of BFO are taken from [F. Xue *et al. Phys. Rev. B* **90**, 220101 (2014)].

TABLE R1: Coefficients of BFO used in the simulation

α_{11}	$-3.580 \times 10^8 \text{ C}^{-2}\text{m}^2\text{N}$	κ_{1111}	$7.840 \times 10^{-11} \text{ N}$
α_{1111}	$3.000 \times 10^8 \text{ C}^{-4}\text{m}^6\text{N}$	κ_{1122}	$-5.138 \times 10^{-9} \text{ N}$
α_{1122}	$1.188 \times 10^8 \text{ C}^{-4}\text{m}^6\text{N}$	κ_{1212}	$4.977 \times 10^{-9} \text{ N}$
β_{11}	$-5.400 \times 10^9 \text{ Nm}^{-2}$	c_{1111}	$2.280 \times 10^{11} \text{ Nm}^{-2}$
β_{1111}	$3.440 \times 10^{10} \text{ Nm}^{-2}$	c_{1122}	$1.280 \times 10^{11} \text{ Nm}^{-2}$
β_{1122}	$6.799 \times 10^{10} \text{ Nm}^{-2}$	c_{1212}	$0.650 \times 10^{11} \text{ Nm}^{-2}$
t_{11}	$4.532 \times 10^9 \text{ C}^{-2}\text{m}^2\text{N}$	λ_{1111}	0.08416
t_{1111}	$2.266 \times 10^9 \text{ C}^{-2}\text{m}^2\text{N}$	λ_{1122}	-0.09200
t_{1122}	$-4.840 \times 10^9 \text{ C}^{-2}\text{m}^2\text{N}$	λ_{1212}	0.3192
g_{1111}	$4.335 \times 10^{-11} \text{ C}^{-2}\text{m}^4\text{N}$	h_{1111}	$0.05700 \text{ C}^{-2}\text{m}^4$
g_{1122}	$-3.400 \times 10^{-12} \text{ C}^{-2}\text{m}^4\text{N}$	h_{1122}	$-0.02000 \text{ C}^{-2}\text{m}^4$
g_{1212}	$3.400 \times 10^{-12} \text{ C}^{-2}\text{m}^4\text{N}$	h_{1212}	$-0.0007300 \text{ C}^{-2}\text{m}^4$

3. The depolarization field : To consider the effect of the depolarization field along the

out-of-plane direction, we calculate the average polarization $\bar{P}_3 = \frac{\sum_{i=1}^n P_3}{n}$, and the

depolarization electric field $E_3 = -\frac{\bar{P}_3}{\varepsilon_b \varepsilon_0} + E_{ex}$, where E_{ex} is the extra electric field

caused by the flexoelectric effect and its magnitude is tuned to obtain the domain structures similar to experiments.

4. *Boundary conditions* : To describe the mechanical boundary conditions of BFO nanoplates, the system consists of three types of materials, i.e., BFO, air, and substrate. BFO possesses nonzero polarization, and the polarization in the air and substrate is zero. The elastic stiffness of the air is zero, and we assume that the elastic stiffness of the substrate is the same as BFO, which guarantees that the bottom interface is constrained while the other five surfaces are stress-free. Temporal evolution of the order parameter is described by the time-dependent Ginzburg-Landau equation, $\partial P_i / \partial t = -L_p (\delta F / \delta P_i)$ and $\partial \theta_i / \partial t = -L_\theta (\delta F / \delta \theta_i)$, which is solved numerically using the semi-implicit Fourier spectral method. Periodic boundary conditions are applied along three directions, and a spectral iterative perturbation method is used to solve the mechanical and electrostatic equilibrium conditions.

This simulation methodology including the model and boundary conditions is described intimately in the Methods section, and newly updated simulation results are reflected in Figs. 4 & 5 and Fig. S2. We are very glad that the updated simulation results can explain more details of experimental observations including the emergence of buffer domains.

Reviewers' Comments:

Reviewer #1 (Remarks to the Author):

The authors have made a thorough revision of the paper, taking account of my suggestions, and also suggestions made by other reviewers. I think the revisions make the paper much better and although some aspects could still be questioned, I believe the results are credible and significant enough for publication in Nature Comms.

Reviewer #2 (Remarks to the Author):

The authors have made substantial changes to the manuscript and nicely addressed all my comments. I can recommend its publication in Nature Communications.

Reviewer #3 (Remarks to the Author):

The authors have satisfactorily addressed my concerns and I now recommend the paper for publication.

Reviewers' comments:

Reviewer #1 (Remarks to the Author):

The authors have made a thorough revision of the paper, taking account of my suggestions, and also suggestions made by other reviewers. I think the revisions make the paper much better and although some aspects could still be questioned, I believe the results are credible and significant enough for publication in Nature Comms.

We thank Reviewer #1 for this supportive evaluation.

Reviewer #2 (Remarks to the Author):

The authors have made substantial changes to the manuscript and nicely addressed all my comments. I can recommend its publication in Nature Communications.

We are grateful to Reviewer #2 for the supportive comment.

Reviewer #3 (Remarks to the Author):

The authors have satisfactorily addressed my concerns and I now recommend the paper for publication.

We appreciate Reviewer #3 for the encouraging assessment.